# The incidence of acute respiratory infection in Indonesian infants and association with vitamin D deficiency

Vicka Oktaria[1,2]*, Margaret Danchin[1], Rina Triasih[2], Yati Soenarto[2], Julie E. Bines[1], Anne-Louise Ponsonby[1], Michael W. Clarke[3], Stephen M. Graham[1]

1 Department of Paediatrics, Murdoch Children's Research Institute, Royal Children's Hospital, University of Melbourne, Melbourne, Australia, 2 Child Health Department, Faculty of Medicine, Public Health, and Nursing, Universitas Gadjah Mada, Yogyakarta, Indonesia, 3 Metabolomics Australia, Centre for Microscopy, Characterisation, and Analysis, and School of Biomedical Sciences, Faculty of Health, and Medical Sciences, The University of Western Australia, Perth, WA, Australia

* vicka.oktaria@ugm.ac.id

## Abstract

### Background

Vitamin D deficiency has been associated with acute respiratory infection (ARI) in early life, but this has not been evaluated in Indonesia. We aimed to determine the incidence of ARI in Indonesian infants, and to evaluate the association with vitamin D deficiency.

### Methods

From 23 December 2015 to 31 December 2017, we conducted a community-based prospective cohort study in Yogyakarta province. We enrolled 422 pregnant women and followed their infants from birth until 12 months of age for ARI episodes. Vitamin D status was measured at birth and at age six months. We performed Cox proportional hazard regression analysis to evaluate the association between vitamin D deficiency and pneumonia incidence.

### Results

At study completion, 95% (400/422) of infants retained with a total of 412 child years of observation (CYO). The incidence of all ARI and of WHO-defined pneumonia was 3.89 (95% CI 3.70–4.08) and 0.25 (95% CI 0.21–0.30) episodes per CYO respectively. Vitamin D deficiency at birth was common (90%, 308/344) and associated with more frequent episodes of ARI non-pneumonia (adjusted odds ratio 4.48, 95% CI:1.04–19.34). Vitamin D status at birth or six months was not associated with subsequent pneumonia incidence, but greater maternal sun exposure during pregnancy was associated with a trend to less frequent ARI and pneumonia in infants.

### Conclusion

ARI, pneumonia, and vitamin D deficiency at birth were common in Indonesian infants. Minimising vitamin D deficiency at birth such as by supplementation of mothers or safe sun

**Data Availability Statement:** The data underlying this study are available on Figshare (DOI: 10.6084/m9.figshare.13726525).

**Funding:** This study was supported by Murdoch Children's Research Institute in the form of funding awarded to SMG, Schlumberger foundation faculty for the future in the form of funding awarded to VO, Indonesia Endowment Fund for Education (LPDP) Ministry of Finance in the form of a grant awarded to VO (20130822080370), the David Bickart Clinician Research Fellowship from the University of Melbourne awarded to MD, Australia-Indonesia Centre (AIC) in the form of a grant awarded to MD and YS (01HSP1MELDancUGM003), and infrastructure funding from the Western Australian State Government, in partnership with the Australian Federal Government, through Bioplatforms Australia and the National Collaborative Research Infrastructure Strategy awarded to MWC. The funders had no role in study design, data collection and analysis, decision to publish, or preparation of the manuscript.

**Competing interests:** The authors have declared that no competing interests exist.

**Abbreviations:** ARI, Acute respiratory infection; CYO, Child years of observation; EBF, Exclusive breastfeeding; HR, Hazard Ratio; IUFD, Intra Uterine Fetal Death; LRTI, Lower Respiratory Tract Infections; OR, Odd Ratio; RCT, Randomised Controlled Trial; UVB, Ultraviolet B; URTI, Upper Respiratory Tract Infections; WHO, World Health Organisation.

exposure during pregnancy has the potential to reduce ARI incidence in infants in this setting.

## Introduction

Acute respiratory infection (ARI) is the leading cause of disease and death in young children (<5 years) globally [1, 2]. ARI incorporates a wide range of respiratory illnesses from mild to life-threatening, that are classified based on location of infection in relation to the vocal cords into: upper respiratory tract infection (URTI) such as rhinitis, nasopharyngitis, tonsillitis, or epiglottitis; and lower respiratory tract infection (LRTI) such as pneumonia, bronchiolitis or croup [3]. Upper respiratory tract infection (URTI) is a very common outpatient presentation associated with low mortality but significant health systems costs [1, 4]. Lower respiratory tract infection (LRTI), specifically pneumonia, is a major cause of hospitalisation and mortality in young children, with infants (<1 year) at highest risk for death.

Indonesia, along with India, Nigeria, Pakistan, and China, contributes to more than half of the estimated 138 million pneumonia cases in young children globally in 2015 [5]. In the same year, the total number of estimated pneumonia episodes in Indonesian children aged younger than five years old were 3,196,000 episodes (2,447,000 to 3,666,000 episodes), with an incidence rate of ~ 300 cases per 1000 population.(1, 3) The total number of estimated deaths reported in that year was 15,250 deaths (9,900 to 20,124 deaths) [2, 5–7]. However, prospective community-based cohort studies that measure actual pneumonia incidence are scarce in high-burden, resource-limited settings such as in Indonesia [8]. Data in Indonesian children are drawn either from prospective cohort studies conducted more than a decade ago prior to the implementation of *Haemophilus influenzae* type b vaccine into the Indonesian national immunisation program in 2013, or from epidemiological modelling studies [2, 9].

The major recognised bacterial pathogens causing pneumonia, influencing global efforts to prevent pneumonia-related deaths through case management and vaccines, have been *Streptococcus pneumoniae* and *Haemophilus influenzae* type b (Hib) [2]. Since the Hib vaccine was included in the routine immunisation schedule in 2013, the prevalence of Hib carriage has been shown to decline to 0% compared to 5% as reported in 1998 [10, 11]. With socioeconomic development and the introduction of bacterial conjugate vaccines, the epidemiology is changing with a relatively larger proportion of ARI due to viruses and a lower mortality attributed to ARI [12]. Nonetheless, there remains huge potential for public health interventions that can further reduce the significant incidence and morbidity due to ARI in infants [13].

ARI risk factors in Indonesia are similar to the global population. These include poverty, indoor pollution, suboptimum exclusive breastfeeding and poor nutritional status [5, 14]. Nutritional deficiency, including both macro and micronutrient deficiencies, is common in Indonesian children [15, 16]. The potential of vitamin D to prevent and/or treat ARI in children has been a focus of recent research [17–19]. Rickets, the most severe clinical manifestation of vitamin D deficiency, has been reported as common in young children with severe pneumonia in some settings [20]. Subclinical vitamin D deficiency (serum vitamin D <50 nmol/L) is far more common than clinical rickets in young children [21–23] and has also been associated with ARI, including increased risk of URTI, LRTI, and LRTI hospitalisation [18, 19, 24]. There is biological plausibility for this association given the immunomodulatory effects of vitamin D deficiency [25]. However, a limited number of studies on vitamin D supplementation in children with ARI have been published to date and evidence of a convincing association

with prevention or treatment of outcomes is still lacking [17, 26]. The World Health Organisation (WHO) has called for more research to guide recommendations for the role of vitamin D supplementation to prevent ARIs [27]. A high prevalence of vitamin D deficiency has been reported in children in South East Asia, particularly in newborns [21, 28, 29]. The prevalence of vitamin D deficiency in Indonesia children aged 2–5 years old was 44% and newborns have been reported at high risk of vitamin D deficiency [29, 30].

We aimed to determine ARI incidence in a cohort of Indonesian infants and the association between vitamin D deficiency at birth and at six months of age with the development of ARI, including of WHO-defined pneumonia. We also evaluated the risk factors for the first episode of pneumonia.

## Materials and methods

A prospective birth cohort study (the Indonesian Pneumonia and vitamin D study—IPAD study) was conducted in nine Primary Healthcare Centres (Jetis, Tegalrejo, Gedongtengen, Gondokusuman 2, Mantrijeron, Gondomanan, Wates, Sentolo, Pengasih 1) and five private practice clinics (Sri Suharti, Suwarti, Sulalita, Sri Suyantiningsih, and Sri Esthini) located in two districts in Yogyakarta province, Indonesia. Yogyakarta province is populated by approximately 3.7 million people (1.4%, 3.7/255 million people of the Indonesian population) with the two study districts, Kota Yogyakarta and Kulon Progo, each having approximately 400,000 people.

### Study procedures

An attending study midwife or doctor recruited mothers during the third trimester of pregnancy at routine antenatal visits. Inclusion criteria included expected delivery in one of the study sites without intention to leave the area within 12 months. Parents who did not provide written informed consent for their child's participation or who did not have phone access were not included. Informed consent was obtained prior to delivery and included the intention to actively follow infants for 12 months, and to collect cord blood at delivery and a venous blood sample at six months of age. Written informed consent was reaffirmed after delivery of a live birth.

Active and passive surveillances were undertaken to capture ARI episodes during the 12-month period of observation. Ten routine follow-up visits within the child's first year, included seven face-to-face visits at ages 1, 2, 3, 4, 6, 9, and 12 months and three phone interviews at ages 5, 7, and 10 months. Routine two-weekly calls were conducted to complement the monthly data collection. At the first visit, we interviewed parents using a structured questionnaire (S1 File) to ascertain demographic data, pregnancy, maternal, and infant sun behaviour, and birth history. During each follow-up visit or routine two-weekly calls, parents were specifically asked about the occurrence of any respiratory symptoms since the previous visit/call. Parents were encouraged to contact the study team within 24 hours of the onset of respiratory symptoms to seek medical advice. A home visit by the study team was performed if the infants reported cough for more than three days to obtain the respiratory rate, assess the severity of the ARI episode, and provide appropriate medical advice regarding the need for referral to the study Primary Healthcare Centre to receive antibiotics for possible pneumonia, according to WHO guidelines [31]. Phone follow-up for ARI episodes were conducted at 3-day intervals until the symptom resolution. Routine medical chart review was conducted at the study sites by the study team. A study booklet was provided to all mothers to record information on any medical services sought for the infant. If infants required hospitalisation for respiratory infections in non-study hospitals, with the parent's permission, the hospital discharge letter was accessed to record diagnosis, clinical management, and outcome.

All study team members completed mandatory training for identification of the signs and case management of pneumonia as per the current WHO classification [20]. Criteria for WHO-defined "pneumonia" are cough with fast breathing or chest indrawing, without danger signs (i.e. cyanosis, hypoxia, unable to drink, or seizure); for "severe pneumonia" with any danger sign present [31]. ARI non-pneumonia episode was defined for respiratory symptoms (including runny nose, fever, sore throat and/or cough) without fast breathing or chest indrawing. A new episode of ARI was defined if there were seven symptom-free days since the previous episode, and of pneumonia after one symptom-free month after the last episode of pneumonia. ARI-received antibiotics was the term used when parents reported that infants had ARI-non-pneumonia diagnosed after antibiotics had been administered, or any ARI with antibiotics administration but information from the medical record or parental reporting was incomplete/insufficient for the study team to classify ARI as pneumonia or non-pneumonia. Maternal sun-exposure during pregnancy or cumulative infant sun-exposure was defined as cumulative composite estimate UVR which was a product of time spent outdoors multiplied by the ambient ultraviolet B during the period of time (per 100 mW/m2/nm/hour).

Sera from cord blood and venous blood were assayed at Metabolomics Australia, Centre for Microscopy, Characterisation, and Analysis, University of Western Australia. The total serum 25-hydroxyvitamin D concentrations were measured using liquid chromatography-tandem mass spectrometry, the current gold standard for vitamin D measurement, which has shown excellent agreement with Center for Disease Control and Prevention's 25-hydroxyvitamin D inaugural Vitamin D standardisation program ($r2 = 0.99$) [32]. The analytical sensitivity (or limit of detection) and the functional sensitivity (or limit of quantitation) of the assay for 25-hydroxyvitamin D3 is 0.5 nM and 2 nmol/L respectively.

Commercial quality controls are assayed at the beginning and end of each batch, with intra-assay Coefficient of variability's (CV's) for all three metabolites <5%, with inter-assay CV's being <10%. Internal standard peak areas are monitored for each batch and are <10% within each run. We defined vitamin D deficiency as a serum 25-hydroxyvitamin D3 level of <50 nmol/L.

## Statistical analysis

All analyses were performed using STATA 15 (Stata Corp, College Station, TX). We aimed to recruit at least 250 infants over the 12-month study period. Allowing for a 10% refusal rate and 10% loss to follow-up prior to the first post-natal visit, this enabled the description of the incidence of pneumonia to within 4–6% based on an incidence rate of 0.1 to 0.5 per person-year (primary objective) based on a two-sided 95% confidence interval.

The time period of observation for each infant was determined as the period in days from the date of birth until the date of the last contact made (final follow-up, prior to lost to follow-up or date of death) to calculate child years of observation (CYO). Cox proportional hazard regression model of survival analysis was performed to explore the association between vitamin D status at birth with the hazard rate of having a first episode of pneumonia in the first and second six months, and the association between vitamin D status at six months and the hazard rate of having the first episode of pneumonia when infants were aged between 6–12 months. Linear and logistic regression were performed to explore the association between vitamin D status and the total number of ARI episodes. Confounding was determined: if the difference between the Odd Ratio (OR) or Hazard Ratio (HR) in the univariate and multivariate models changed by more than 10% or based on a priori knowledge. A sensitivity analysis using a lower cut off vitamin D level of <25 nmol/L (severe vitamin D deficiency) and a higher cut off vitamin D level of < 75 nmol/L was performed. A p-value <0.05 was considered statistically significant.

## Ethics

We obtained ethical approval for the study from the Medical and Health Research Ethics Committee of Universitas Gadjah Mada Yogyakarta, Indonesia (ethics approval number KE/FK/935/EC/2015, July 2015) and the University of Melbourne Human Research Committee (ethics approval number 1544817, October 2015).

## Results

Between 23 December 2015 and 31 December 2017, we recruited 422 infants with 95% (400/422) of infants retained for the full twelve months of follow-up. Of recruited infants, 82% (348/422) had cord blood samples and 60% (255/422) had venous blood samples at six months of age (Fig 1).

The demographics characteristics of infants with vitamin D samples at birth and six months of age were representative of the entire cohort (Table 1). The mean serum vitamin D concentrations at birth and at six months were 30 nmol/L ± 14 nmol/L and 77 ± 26 nmol/L, respectively. The prevalence of vitamin D deficiency was 90% (308/344) at birth and 13% (33/255) at six months (± 4 weeks).

An analysis of the determinants of vitamin D deficiency at birth and at 6 months has been reported elsewhere [29]. Mothers who spent two hours or more per day outside were less likely to have a newborn with vitamin D deficiency compared to those who spent only 15 minutes or less in the sun [29].

Overall, there were 1,601 ARI episodes identified within 412 CYO. The WHO criteria for pneumonia and severe pneumonia were met for 96 and 7 of the ARI episodes respectively, and 8.7% (9/103) of the pneumonia episodes were hospitalised. Almost all (96%) infants had at least one episode of ARI non-pneumonia (Table 2). The incidence rate for all ARI was 3.89 (95% CI: 3.70–4.08) and for pneumonia was 0.25 (95% CI: 0.21–0.30) episodes per CYO (Table 3). The highest incidence rate for all ARI and pneumonia was in the 9 to <12-month age group and the lowest incidence rate was in infants <3 months of age. The duration of illness to complete symptom resolution for non-pneumonia and pneumonia ARI was 7 days (4–9 days) and 15 days (11–26 days), respectively. Of pneumonia episodes, 6.8% (7/103) presented with wheezing and 2.9% (3/103) were hypoxemic.

The incidence rate for the first episode of pneumonia within the first and second six months was not different between those with vitamin D deficiency and vitamin D sufficiency at birth (level ≥ 50 nmol/L) (Table 4). Similarly, vitamin D status at six months of age was not associated with the incidence rate of first pneumonia episode that occurred between six and 12 months of age (Table 5). Kaplan-Meier curves of pneumonia-free survival by (a) vitamin D status at birth and by (b) vitamin D status at 6 months of age are plotted in Fig 2. A sensitivity analysis using a lower cut off vitamin D level of <25 nmol/L did not alter the results. Male-sex and passive exposure to paternal smokers were associated with a higher rate of pneumonia. Maternal sun exposure during pregnancy, but not infant sun exposure from birth to six months of age, was associated with a reduced risk of pneumonia (Table 5) and less frequent ARI non-pneumonia during infancy (adjusted OR 0.78; 95% CI: 0.60–0.98, result not shown in the table). There were weak correlations between maternal and infant sun exposure: Spearman's correlation coefficients r 0.12, $P$ 0.02 for weekday and weekend exposure.

The risk of having six or more episodes of ARI non-pneumonia was higher in infants with vitamin D deficiency at birth, even after adjustment for paternal smoking status, low birth weight, overcrowding household, and exclusive breastfeeding status (adjusted OR 4.48; 95% CI: 1.04–19.34, $P$ 0.04). Other risk factors for having one additional episode of

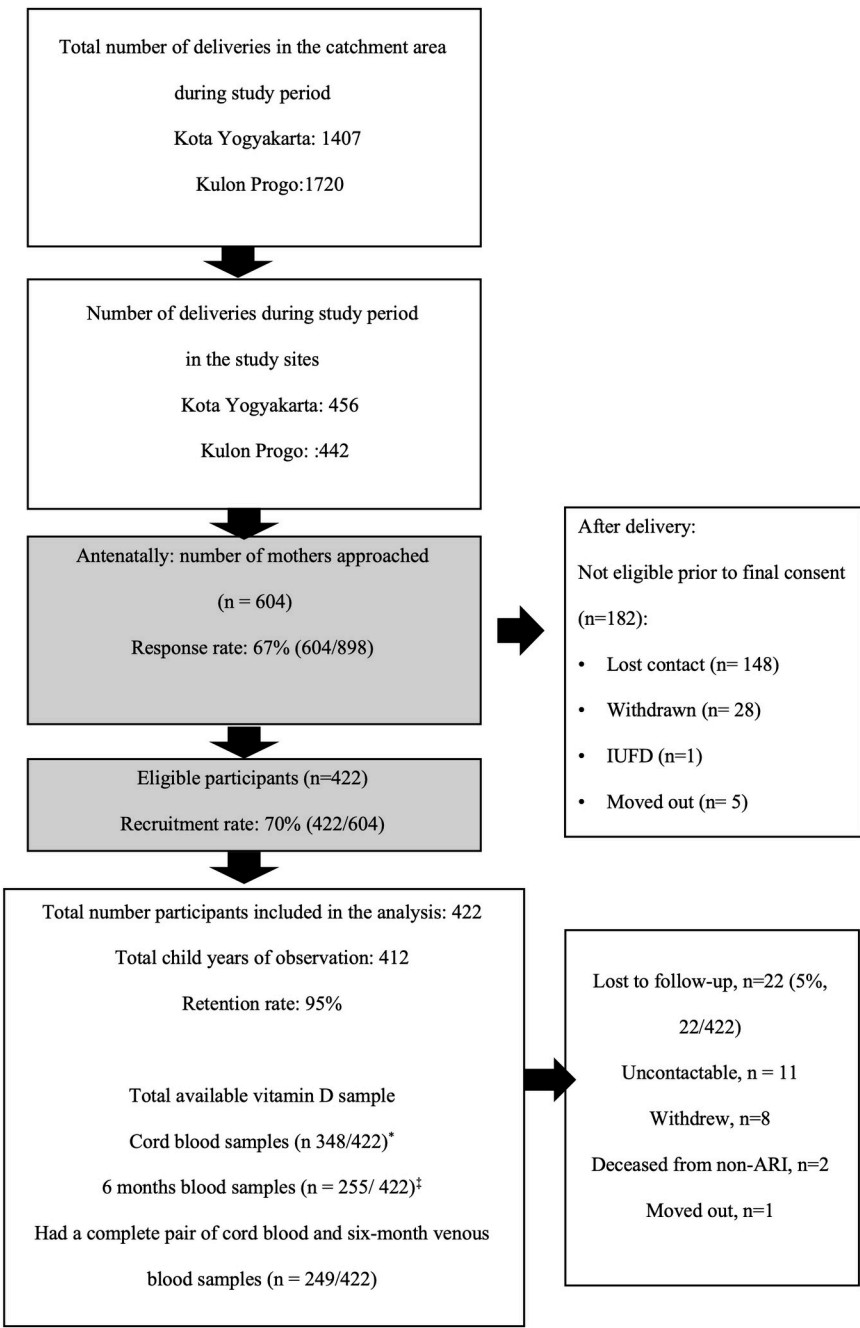

**Fig 1. A flow chart depicts infants flow and vitamin D sample availability.** *74 cord blood samples were not collected as the mother was referred to hospital due to complication or post-dates (n = 47); or midwife missed the blood collection (n = 27); ‡ 95 Venous blood samples were not collected: no cord blood samples (n = 60); lost to follow-up before age turned to 6 months old (n = 1; parents refused for blood collection (n = 10; parent refused, LTFU (n = 2), venous blood was collected at ≥7 months of age (n = 22); ‡2 Venous blood without cord blood samples at birth were collected due to infants developed pneumonia later after birth; IUFD: Intra Uterine Fetal Death.

non-pneumonia ARI were household crowding (adjusted beta-coefficient 0.12, 95% CI: 0.0–0.23 per additional one person); and preterm born (adjusted beta-coefficient 1.55, 95% CI: 0.23–2.88).

**Table 1. Demographic characteristics and ARI outcomes of infants by vitamin D sample availability.**

| | | Entire cohort N = 422 N (%) | Group 1: Infants with cord blood N = 344 (82% of total cohort) | Group 2: Infants with venous blood at 6 months of age N = 255 (60% of total cohort) |
|---|---|---|---|---|
| **Demographic characteristics** | | | | |
| | Male infants–n (%) | 209 (50%) | 168 (49%) | 128 (50%) |
| | Birth weight in grams–(median, IQR) | 3100 (2900, 3400) | 3100 (2900, 3350) | 3100 (2900, 3400) |
| | Premature birth | 8 (2%) | 6 (2%) | 5 (2%) |
| | Low birth weight | 17 (4%) | 14 (4%) | 12 (5%) |
| | Maternal age in years–(median, IQR) | 29 (24, 33) | 29 (24, 33) | 29 (25, 33) |
| | Paternal smoking–n (%) | 204 (49%) | 169 (49%) | 130 (51%) |
| | Number of people living with infants (median, IQR), | 4 (3–5) | 4 (3–5) | 4 (3–5) |
| | Family income per month–n (%) | | | |
| | < IDR 1000k | 207 (49%) | 175 (51%) | 140 (55%) |
| | IDR 1000k – 5000k | 200 (48%) | 156 (46%) | 107 (42%) |
| | IDR > 5000k | 14 (3%) | 12 (4%) | 8 (3%) |
| | Exclusive breastfeeding–n (%) | 228 (56%) | 191 (58%) | 155 (61%) |
| **ARI outcomes** | | | | |
| | Total ARI episodes,—(mean ± SD) | 3.80 (± 2) | 3.69 (± 2) | 3.96 (± 2) |
| | Infants with pneumonia–n (%) | 89 (21%) | 76 (22%) | 66 (26%) |

## Discussion

We report a high incidence of ARI and pneumonia in Indonesian infants. One in four had an episode of pneumonia in the first year of life, consistent with recent estimates in low- and

**Table 2. The total number of ARI episodes in the study cohort.**

| WHO-defined pneumonia | No. infants (%) |
|---|---|
| 0 episode | 332 (79%) |
| 1 episode | 76 (18%) |
| 2 episodes | 12 (2·5%) |
| 3 episodes | 1 (0·5%) |
| **ARI non-pneumonia** | **No. infants (%)** |
| Median (IQR) | 5 (3–6) |
| 0 episode | 23 (5%) |
| 1 episode | 41(10%) |
| 2 episodes | 59 (14%) |
| 3 episodes | 89 (21%) |
| 4 episodes | 80 (19%) |
| 5 episodes | 63 (15%) |
| 6 episodes | 39 (9%) |
| 7 episodes | 15 (4%) |
| 8 episodes | 10 (2%) |
| 9 episodes | 2 (1%) |

ARI non-pneumonia defined respiratory symptoms such as cough but without fast breathing or chest indrawing, including rhinitis, rhino pharyngitis but the diagnosis was not further specified.

**Table 3. The incidence of ARI by age.**

| | Total CYO | Incidence rates episodes per CYO (95% Cis) Incidence rates per CYO (95% Cis) | | | | | |
| | | ALL ARI | ARI non-pneumonia | ARI received antibiotics | WHO-defined pneumonia | WHO-defined severe pneumonia | Hospitalised pneumonia |
|---|---|---|---|---|---|---|---|
| **All** | 412 | 3.89 (3.70–4.08) | 3.63 (3.46–3.83) | 0.62 (0.55–0.70) | 0.25 (0.21–0.30) | 0.02 (0.01–0.04) | 0.02 (0.01–0.04) |
| **Per 3 months age group** | | | | | | | |
| 0–2 months | 104 | 1.87 (1.62–2.14) | 1.78 (1.54–2.06) | 0.14 (0.08–0.23) | 0.08 (0.04–0.15) | 0.01 (0.001–0.07) | 0.02 (0.005–0.08) |
| 3–5 months | 102 | 3.94 (3.57–4.34) | 3.68 (3.33–4.07) | 0.32 (0.23–0.46) | 0.26 (0.17–0.38) | n/a | 0.02 (0.005–0.08) |
| 6–8 months | 101 | 4.53 (4.13–4.97) | 4.25 (3.87–4.67) | 0.82 (0.66–1.01) | 0.28 (0.19–0.40) | 0.03 (0.01–0.09) | 0.05 (0.02–0.12) |
| 9–<12 months | 105 | 5.22 (4.81–5.68) | 4.84 (4.44–5.28) | 1.19 (1.00–1.41) | 0.39 (0.29–0.53) | 0.03 (0.01–0.09) | n/a |
| **WHO age group** | | | | | | | |
| < 2 months | 70 | 1.70 (1.42–2.03) | 1.61 (1.34–1.94) | 0.17 (0.10–0.30) | 0.09 (0.04–0.19) | 0.01(0.002–0.10) | 0.03 (0.007–0.12) |
| 2–<12 months | 342 | 4.33 (4.12–4.56) | 4.05 (3.84–4.27) | 0.71 (0.62–0.80) | 0.28 (0.23–0.35) | 0.02 (0.008–0.04) | 0.02 (0.009–0.04) |

CYO = child year observation.

ARI-received antibiotics was the term used when parents reported that infants had ARI-non-pneumonia diagnosed (above classification) after antibiotics had been administered, or any ARI with antibiotics administration but information from medical record or parental reporting was incomplete/insufficient for study team to classify ARI as pneumonia or non-pneumonia.

middle-income countries [5, 33]. Vitamin D deficiency at birth was associated with more frequent episodes of non-pneumonia ARI but not with pneumonia incidence or severity.

The incidence of pneumonia of 0.25 episodes per CYO in our study is comparable to other studies in similar settings. A study in Indonesian infants done nearly two decades ago prior to implementation of the Hib vaccine recorded 0.21 pneumonia episodes per CYO using only passive surveillance [34]. A recent South African birth cohort study of 697 newborns that used active surveillance similar to our study reported 0.27 (95% CI 0.23–0.32) pneumonia episodes during infancy [34, 35]. The reported incidence of all ARI differs more widely. A birth cohort study of 2459 newborns from Vietnam in 2010 that used similar surveillance methods, included urban and semi-rural settings and had a comparable risk factor profile to our study reported fewer episodes of ARI (0.5–2 per CYO) by 12 months of age, but also reported a lower incidence of ARI in infants less than six months of age compared to older infants [33]. Comparisons between studies are often challenging to interpret given that the differences in time periods or epidemiological settings are likely to affect a range of risk factors for ARI in infants which strongly affect the incidence of ARI in the settings.

Known risk factors for pneumonia of male-sex and passive smoking exposure were confirmed in our cohort, consistent with previous studies [35, 36]. Vitamin D deficiency was common at birth or at six months of age but was not associated with subsequent incidence or severity of pneumonia. A recent study from Bangladesh also reported no association between vitamin D status at birth or in infancy and the subsequent development of pneumonia or LRTI [37]. In contrast, an association was reported in a smaller study of young children in Saudi Arabia (adjusted OR 1.08, 95% CI: 1.05–1.10, $P <0.001$), but this was a retrospective study with passive case-ascertainment that only collected data at the end of the follow-up period requiring medical record review [19]. Active case-ascertainment with frequent follow-up was a strength of our study to reduce the risk of recall bias and misclassification [8, 38]. Large variations in the methodological approaches in observational studies that have previously evaluated

**Table 4. Hazard rates and 95% confidence intervals for possible risk factors associated with first episode of pneumonia during infancy by age group.**

| | Pneumonia episode at 0–5 months | | | Pneumonia episode at 6–11 months | | | Pneumonia episode during infancy | | |
|---|---|---|---|---|---|---|---|---|---|
| | Cases | CYO | Hazard Rate (95% CI) | Cases | CYO | Hazard rate (95% CI) | Cases | CYO | Hazard rate (95% CI) |
| **Sex** | | | | | | | | | |
| Female | 9 | 98 | 0.09 (0.05–0.18) | 22 | 93 | 0.24 (0.15–0.36) | 31 | 190 | 0.16 (0.11–0.23) |
| Male | 23 | 96 | 0.24 (0.16–0.36) | 35 | 82 | 0.43 (0.31–0.59) | 58 | 178 | 0.33 (0.25–0.42) |
| **Prematurity** | | | | | | | | | |
| At term | 30 | 188 | 0.16 (0.11–0.23) | 54 | 169 | 0.32 (0.25–0.42) | 84 | 356 | 0.24 (0.19–0.29) |
| Less than 37 weeks | 0 | 0 | 0 | 3 | 4 | 0.74 (0.24–2.30) | 3 | 8 | 0.37 (0.12–1.16) |
| **Low birth weight** | | | | | | | | | |
| Normal | 30 | 182 | 0.17 (0.12–0.24) | 56 | 163 | 0.34 (0.27–0.45) | 86 | 345 | 0.24 (0.20–0.30) |
| Low | 0 | 8·5 | 0 | 1 | 9 | 0.11 (0.02–0.80) | 1 | 17 | 0.58 (0.01–0.41) |
| **EBF** | | | | | | | | | |
| Yes | 20 | 107 | 0.19 (0.12–0.29) | 27 | 99 | 0.27 (0.19–0.40) | 47 | 207 | 0.23 (0.17–0.30) |
| No | 12 | 84 | 0.14 (0.08–0.25) | 30 | 75 | 0.40 (0.28–0.57) | 42 | 160 | 0.26 (0.19–0.36) |
| **Number of people living in the house** | | | | | | | | | |
| < 7 people | 24 | 171 | 0.14 (0.09–0.21) | 50 | 156 | 0.32 (0.24–0.42) | 74 | 327 | 0.23 (0.18–0.28) |
| ≥ 7 people | 8 | 23 | 0.35 (0.17–0.69) | 7 | 19 | 0.38 (0.18–0.79) | 15 | 42 | 0.36 (0.22–0.60) |
| **Mother sun exposure for the entire pregnancy** | | | | | | | | | |
| Weekday | | | | | | | | | |
| (per 100 mW/m2/nm/hour). Interquartile range | | | | | | | | | |
| Quartile 1 (91–174) | 9 | 46 | 0.20 (0.10–0.38) | 19 | 39 | 0.49 (0.31–0.76) | 28 | 85 | 0.33 (0.23–0.48) |
| Quartile 2 (183–342) | 6 | 47 | 0.13 (0.06–0.29) | 16 | 43 | 0.37 (0.23–0.61) | 22 | 90 | 0.25 (0.16–0.37) |
| Quartile 3 (363–379) | 10 | 42 | 0.24 (0.13–0.44) | 12 | 36 | 0.33 (0.19–0.58) | 22 | 78 | 0.28 (0.19–0.43) |
| Quartile 4 (743–1134) | 5 | 45 | 0.11 (0.05–0.27) | 9 | 43 | 0.21 (0.11–0.41) | 14 | 88 | 0.16 (0.09–0.27) |
| Weekend | | | | | | | | | |
| (per 100 mW/m2/nm/hour). Interquartile range | | | | | | | | | |
| Quartile 1 (91–174) | 9 | 50 | 0.18 (0.09–0.35) | 20 | 43 | 0.47 (0.30–0.72) | 29 | 92 | 0.31 (0.22–0.45) |
| Quartile 2 (183–192) | 6 | 44 | 0.14 (0.06–0.30) | 13 | 41 | 0.31 (0.18–0.54) | 19 | 86 | 0.22 (0.14–0.35) |
| Quartile 3 (356–378) | 9 | 47 | 0.19 (0.10–0.37) | 16 | 41 | 0.39 (0.24–0.64) | 25 | 88 | 0.29 (0.19–0.42) |
| Quartile 4 (726–1134) | 6 | 49 | 0.12 (0.05–0.27) | 8 | 46 | 0.17 (0.09–0.35) | 14 | 96 | 0.15 (0.09–0.25) |
| **Infant cumulative sun exposure during 0–5 months of age[a]** | | | | | | | | | |
| Weekday | | | | | | | | | |
| (per 100 mW/m2/nm/hour). Interquartile range | | | | | | | | | |
| Quartile 1 (196–256) | | NA | | 14 | 44 | 0.32 (0.19–0.53) | | NA | |
| Quartile 2 (303–350) | | | | 16 | 42 | 0.38 (0.23–0.63) | | | |
| Quartile 3 (394–459) | | | | 13 | 43 | 0.30 (0.18–0.52) | | | |
| Quartile 4 (559–806) | | | | 12 | 41 | 0.29 (0.17–0.51) | | | |
| Weekend | | | | | | | | | |
| (per 100 mW/m2/nm/hour). Interquartile range | | | | | | | | | |
| Quartile 1 (200–260) | | NA | | 16 | 40 | 0.40 (0.25–0.66) | | NA | |
| Quartile 2 (303–351) | | | | 12 | 40 | 0.30 (0.17–0.54) | | | |
| Quartile 3 (396–465) | | | | 13 | 39 | 0.33 (0.19–0.57) | | | |
| Quartile 4 (567–835) | | | | 11 | 38 | 0.29 (0.16–0.52) | | | |
| **Father smokes everyday** | | | | | | | | | |

(*Continued*)

**Table 4.** (*Continued*)

| | | Pneumonia episode at 0–5 months | | | Pneumonia episode at 6–11 months | | | Pneumonia episode during infancy | | |
|---|---|---|---|---|---|---|---|---|---|---|
| | | Cases | CYO | Hazard Rate (95% CI) | Cases | CYO | Hazard rate (95% CI) | Cases | CYO | Hazard rate (95% CI) |
| | No | 12 | 94 | 0.13 (0.07–0.22) | 22 | 88 | 0.25 (0.17–0.38) | 34 | 182 | 0.19 (0.13–0.26) |
| | Yes | 20 | 99 | 0.20 (0.13–0.31) | 34 | 87 | 0.39 (0.28–0.55) | 54 | 186 | 0.29 (0.22–0.37) |
| **BCG vaccination** | | | | | | | | | | |
| | Yes | 31 | 184 | 0.17 (0.12–0.24) | 55 | 167 | 0.33 (0.25–0.43) | 86 | 351 | 0.25 (0.20–0.30) |
| | No | 1 | 10 | 0.10 (0.01–0.71) | 2 | 8 | 0.25 (0.06–1.00) | 3 | 18 | 0.17 (0.05–0.53) |
| **Cord blood vitamin D status** | | | | | | | | | | |
| | ≥ 50 nmol/L | 4 | 26 | 0.25 (0.09–0.66) | 5 | 14 | 0.36 (0.15–0.86) | 9 | 30 | 0.30 (0.16–0.57) |
| | < 50 nmol/L | 25 | 140 | 0.18 (0.12–0.26) | 42 | 126 | 0.33 (0.25–0.45) | 67 | 266 | 0.25 (0.20–0.32) |
| **Vitamin D status at 6 months** | | | | | | | | | | |
| | ≥ 50 nmol/L | | | NA | 32 | 91 | 0.35 (0.25–0.50) | | | NA |
| | < 50 nmol/L | | | | 7 | 14 | 0.51 (0.24–1.07) | | | |

Cases = number of infants had first episode of pneumonia.

CYO = child year of observation.

EBF = Exclusive breastfeeding.

Low birth weight (below 2500 grams) versus normal birth weight (2500–4000 grams).

Maternal sun exposure during pregnancy was defined as cumulative composite estimate UVR exposure during pregnancy which was product of time spent outdoors multiplied by the ambient ultraviolet B during the period of time (per 100 mW/m2/nm/hour).

Maternal cumulative composite weekday: quartile 1 (91–174 per 100 mW/m2/nm/hour); quartile 2 (183–342 per 100 mW/m2/nm/hour); quartile 3 (363–379 per 100 mW/m2/nm/hour); and quartile 4 (743–1134 per 100 mW/m2/nm/hour).

Maternal cumulative composite weekend: quartile 1 (91–174 per 100 mW/m2/nm/hour); quartile 2 (183–192 per 100 mW/m2/nm/hour); quartile 3 (356–378 per 100 mW/m2/nm/hour); and quartile 4 (726–1134 per 100 mW/m2/nm/hour).

Infant cumulative composite weekday: quartile 1 (196–256 per 100 mW/m2/nm/hour); quartile 2 (303–350 per 100 mW/m2/nm/hour); quartile 3 (394–459 per 100 mW/m2/nm/hour); and quartile 4 (559–806 per 100 mW/m2/nm/hour).

Infant cumulative composite weekend: quartile 1 (200–260 per 100 mW/m2/nm/hour); quartile 2 (303–351 per 100 mW/m2/nm/hour); quartile 3 (396–465 per 100 mW/m2/nm/hour); and quartile 4 (567–835 per 100 mW/m2/nm/hour).

NA = not applicable.

[a] The association between infant cumulative sun exposure and first episode of pneumonia was calculated on the basis of the temporal sequence of exposure before outcome which was not applicable to the association between infant cumulative sun exposure between 0–6 months of age and first episode of pneumonia between 0–6 months of age.

the association between vitamin D deficiency and ARI, including pneumonia, challenges the interpretation of study results [19, 37–40].

Vitamin D deficiency at birth was associated with higher frequency, i.e. six or more episodes, of URTI. However, the incidence of ARI was highest after 6 months of age when the prevalence of vitamin D deficiency was much lower when compared to that observed at birth. A major limitation of interpretation of findings of an association between vitamin D deficiency and disease incidence from observational cohort studies such as our study is the potential for confounding, especially for ARI which has a wide range of potential aetiologies and epidemiological risk factors. Randomised, placebo-controlled vitamin D supplementation trials in populations with a high prevalence of vitamin D deficiency are likely to provide stronger evidence of association if it exists while also assessing the potential benefit for an intervention. Our finding is consistent with findings from a meta-analysis in a wider age range (birth to 95 years of age) that reported that vitamin D supplementation prevented at least one episode of unspecified ARI (adjusted OR 0.88, 95% CI: 0.81–0.96) but not LRTI (adjusted OR 0.96, 95% CI

**Table 5. Cox proportional hazard ratio estimates and 95% confidence intervals for possible risk factors associated with first episode of pneumonia by age group in univariate and multivariate model.**

| 7 | | Hazard Ratio (95% CI) | | |
|---|---|---|---|---|
| | | Pneumonia onset 0–5 months | Pneumonia onset 6–11 months | Pneumonia onset in the first postnatal year |
| Male- vs. female-sex [a] | Univariate | **2.63 (1.21–5.68)** | **1.79 (1.05–3.06)** | **2.04 (1.32–3.15)** |
| | | **P 0.01** | **P 0.01** | **P 0.01** |
| | Multivariate | **2.91 (1.28–6.60)** | **1.82 (1.06–3.15)** | **2.12 (1.35–3.33)** |
| | | **P 0.01** | **P 0.03** | **P 0.001** |
| Non EBF vs. EBF [b] | Univariate | 0.76 (0.37–1.55) | 1.50 (0.89–2.52) | 1.18 (0.78–1.79) |
| | | P 0.45 | P 0.13 | P 0.44 |
| | Multivariate | 0.59 (0.27–1.27) | 1.44 (0.85–2.45) | 1.07 (0.70–1.65) |
| | | P 0.18 | P 0.18 | P 0.75 |
| ≥ 7 people vs. < 7 people lived in the house [c] | Univariate | **2.51 (1.13–5.60)** | 1.17 (0.53–2.59) | 1.63 (0.94–2.85) |
| | | **P 0.02** | P 0.69 | P 0.08 |
| | Multivariate | **2.64 (1.17–5.96)** | 1.13 (0.51–2.51) | 1.61 (0.92–2.81) |
| | | **P 0.02** | P 0.76 | P 0.10 |
| Father smokes everyday vs. non-smoker father [d] | Univariate | 1.58 (0.77–3.24) | 1.56 (0.91–2.67) | **1.57 (1.02–2.41)** |
| | | P 0.21 | P 0.11 | **P 0.04** |
| | Multivariate | 1.87 (0.87–3.99) | 1.59 (0.92–2.73) | **1.67 (1.07–2.60)** |
| | | P 0.11 | P 0.10 | **P 0.02** |
| Cord blood vitamin D status < 50 nmol/L vs. ≥ 50 nmol/L [e] | Univariate | 0.71 (0.25–2.04) | 0.91 (0.36–2.30) | 0.82 (0.41–1.65) |
| | | P 0.53 | P 0.84 | P 0.58 |
| | Multivariate | 0.50 (0.17–1.47) | 0.84 (0.33–2.13) | 0.71 (0.35–1.44) |
| | | P 0.21 | P 0.71 | P 0.34 |
| Vitamin D status at 6 months < 50 nmol/L vs. ≥ 50 nmol/L [e] | Univariate | NA | 1.53 (0.67–3.47) | NA |
| | | | P 0.31 | |
| | Multivariate | | 1.49 (0.63–3.52) | |
| | | | P 0.36 | |
| Mother sun exposure for the entire pregnancy [f] | | | | |
| • Weekday | | | | |
| ◦ Q2 vs. Q1 | Univariate | 0.64 (0.23–1.81) | 0.75 (0.39–1.47) | 0.72 (0.41–1.26) |
| | | P 0.40 | P 0.41 | P 0.25 |
| ◦ Q3 vs. Q1 | Univariate | 1.21 (0.49–2.97) | 0.69 (0.33–1.41) | 0.85 (0.49–1.49) |
| | | P 0.68 | P 0.31 | P 0.58 |
| ◦ Q4 vs. Q1 | Univariate | 0.56 (0.19–1.66) | **0.44 (0.20–0.97)** | **0.48 (0.25–0.90)** |
| | | P 0.29 | **P 0.04** | **P 0.02** |
| Trend per 1 increase of quartile | Univariate | 0.91 (0.66–1.25) | **0.78 (0.61–0.99)** | 0.82 (0.68–1.00) |
| | | P 0.57 | **P 0.04** | P 0.05 |
| ◦ Q2 vs. Q1 | Multivariate | 0.62 (0.22–1.74) | 0.75 (0.38–1.48) | 0.70 (0.40–1.24) |
| | | P 0.36 | P 0.40 | P 0.22 |
| ◦ Q3 vs. Q1 | Multivariate | 1.27 (0.51–3.15) | 0.68 (0.33–1.42) | 0.85 (0.48–1.49) |
| | | P 0.61 | P 0.31 | P 0.57 |
| ◦ Q4 vs. Q1 | Multivariate | 0.55 (0.18–1.65) | 0.45 (0.20–1.00) | **0.47 (0.25–0.90)** |
| | | P 0.28 | P 0.05 | **P 0.02** |
| Trend per 1 increase of quartile | Multivariate | 0.91 (0.66–1.26) | 0.78 (0.61–1.00) | 0.82 (0.68–1.00) |
| | | P 0.58 | P 0.05 | P 0.05 |
| • Weekend | | | | |
| ◦ Q2 vs. Q1 | Univariate | 0.74 (0.26–2.09) | 0.65 (0.32–1.31) | 0.68 (0.38–1.22) |
| | | P 0.57 | P 0.23 | P 0.19 |
| ◦ Q3 vs. Q1 | Univariate | 1.06 (0.42–2.67) | 0.84 (0.44–1.62) | 0.91 (0.53–1.55) |

*(Continued)*

**Table 5.** (Continued)

| 7 | | Hazard Ratio (95% CI) | | |
| --- | --- | --- | --- | --- |
| | | Pneumonia onset 0–5 months | Pneumonia onset 6–11 months | Pneumonia onset in the first postnatal year |
| | | P 0.90 | P 0.61 | P 0.73 |
| ◦ Q4 vs. Q1 | Univariate | 0.67 (0.24–1.88) | **0.37 (0.16–0.84)** | **0.46 (0.24–0.87)** |
| | | P 0.45 | **P 0.02** | **P 0.02** |
| Trend per 1 increase of quartile | Univariate | 0.92 (0.67–1.26) | **0.78 (0.62–0.99)** | **0.83 (0.69–1.00)** |
| | | P 0.62 | **P 0.05** | **P 0.05** |
| ◦ Q2 vs. Q1 | Multivariate | 0.73 (0.26–2.06) | 0.63 (0.31–1.29) | 0.66 (0.37–1.19) |
| | | P 0.55 | P 0.21 | P 0.16 |
| ◦ Q3 vs. Q1 | Multivariate | 1.15 (0.45–2.92) | 0.88 (0.45–1.71) | 0.94 (0.55–1.61) |
| | | P 0.77 | P 0.70 | P 0.82 |
| ◦ Q4 vs. Q1 | Multivariate | 0.70 (0.25–1.97) | **0.37 (0.16–0.85)** | **0.46 (0.24–0.87)** |
| | | P 0.50 | **P 0.02** | **P 0.02** |
| Trend per 1 increase of quartile | Multivariate | 0.94 (0.69–1.30) | **0.79 (0.62–1.00)** | 0.83 (0.69–1.01) |
| | | P 0.72 | **P 0.05** | P 0.06 |
| Infant cumulative sun exposure between 0–6 months of age[f,g] | | | | |
| • Weekday | | NA | | NA |
| ◦ Q2 vs. Q1 | Univariate | | 1.18 (0.58–2.43) | |
| | | | P 0.65 | |
| ◦ Q3 vs. Q1 | Univariate | | 0.94 (0.44–2.00) | |
| | | | P 0.87 | |
| ◦ Q4 vs. Q1 | Univariate | | 0.93 (0.43–2.00) | |
| | | | P 0.85 | |
| Trend per 1 increase of quartile | Univariate | | 0.96 (0.75–1.21) | |
| | | | P 0.71 | |
| ◦ Q2 vs. Q1 | Multivariate | | 1.16 (0.57–2.39) | |
| | | | P 0.68 | |
| ◦ Q3 vs. Q1 | Multivariate | | 0.88 (0.40–1.91) | |
| | | | P 0.74 | |
| ◦ Q4 vs. Q1 | Multivariate | | 0.90 (0.42–1.96) | |
| | | | P 0.80 | |
| Trend per 1 increase of quartile | Multivariate | | 0.94 (0.74–1.20) | |
| | | | P 0.64 | |
| • Weekend | | | | |
| ◦ Q2 vs. Q1 | Univariate | | 0.73 (0.34–1.55) | |
| | | | P 0.41 | |
| ◦ Q3 vs. Q1 | Univariate | | 0.80 (0.38–1.66) | |
| | | | P 0.55 | |
| ◦ Q4 vs. Q1 | Univariate | | 0.72 (0.33–1.54) | |
| | | | P 0.39 | |
| Trend per 1 increase of quartile | Univariate | | 0.91 (0.71–1.16) | |
| | | | P 0.44 | |
| ◦ Q2 vs. Q1 | Multivariate | | 0.71 (0.33–1.51) | |
| | | | P 0.38 | |
| ◦ Q3 vs. Q1 | Multivariate | | 0.72 (0.33–1.55) | |
| | | | P 0.40 | |
| ◦ Q4 vs. Q1 | Multivariate | | 0.72 (0.33–1.54) | |
| | | | P 0.39 | |
| Trend per 1 increase of quartile | Multivariate | | 0.90 (0.70–1.15) | |

(Continued)

**Table 5.** (Continued)

| 7 | | Hazard Ratio (95% CI) | | |
|---|---|---|---|---|
| | | Pneumonia onset | Pneumonia onset | Pneumonia onset in the |
| | | 0–5 months | 6–11 months | first postnatal year |
| | | | *P* 0.40 | |

[a] Adjusted for paternal smoking, overcrowded, low birth weight and EBF.

[b] Adjusted for paternal smoking, overcrowded, and low birth weight.

[c] Adjusted for paternal smoking, low birth weight, and EBF.

[d] Adjusted for baby sex, overcrowded, EBF, and low birth weight.

[e] Adjusted for birth weight, EBF, paternal smoking, crowding, infants skin type, and month of blood collection.

[f] Adjusted for paternal smoking, overcrowded, and EBF.

Maternal sun exposure during pregnancy was defined as cumulative composite estimate UVR exposure during pregnancy which was product of time spent outdoors multiplied by the ambient ultraviolet B during the period of time (per 100 mW/m2/nm/hour).

Maternal cumulative composite weekday: quartile 1 (91–174 per 100 mW/m2/nm/hour); quartile 2 (183–342 per 100 mW/m2/nm/hour); quartile 3 (363–379 per 100 mW/m2/nm/hour); and quartile 4 (743–1134 per 100 mW/m2/nm/hour).

Maternal cumulative composite weekend: quartile 1 (91–174 per 100 mW/m2/nm/hour); quartile 2 (183–192 per 100 mW/m2/nm/hour); quartile 3 (356–378 per 100 mW/m2/nm/hour); and quartile 4 (726–1134 per 100 mW/m2/nm/hour).

Infant cumulative composite weekday: quartile 1 (196–256 per 100 mW/m2/nm/hour); quartile 2 (303–350 per 100 mW/m2/nm/hour); quartile 3 (394–459 per 100 mW/m2/nm/hour); and quartile 4 (559–806 per 100 mW/m2/nm/hour).

Infant cumulative composite weekend: quartile 1 (200–260 per 100 mW/m2/nm/hour); quartile 2 (303–351 per 100 mW/m2/nm/hour); quartile 3 (396–465 per 100 mW/m2/nm/hour); and quartile 4 (567–835 per 100 mW/m2/nm/hour).

NA = not applicable.

[g] The association between infant cumulative sun exposure and first episode of pneumonia was calculated on the basis of the temporal sequence of exposure before outcome which was not applicable to the association between infant cumulative sun exposure between 0–6 months of age and first episode of pneumonia between 0–6 months of age.

0.83–1.10) [41]. Two RCTs conducted in infants and young children in Afghanistan reported that a bolus dose of 100,000 U vitamin D prevented recurrence of pneumonia within 90 days after hospitalisation but did not prevent pneumonia in healthy infants [42, 43]. There is a broad hierarchy of ARI outcomes and interventional doses in the published RCTs that might

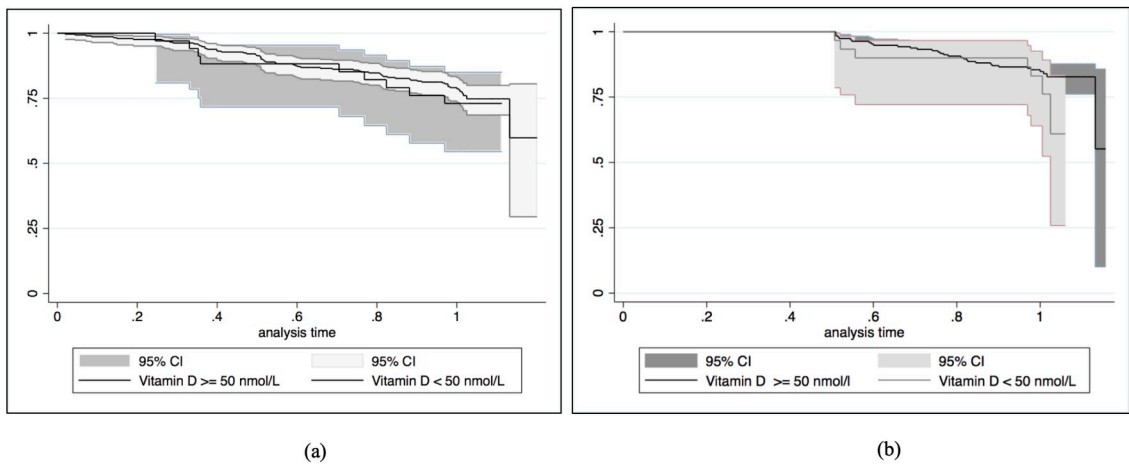

**Fig 2. Time to develop first episode of pneumonia by vitamin D status at birth and 6 months of age.** (a) Kaplan-Meier survival curve of the first episodes of pneumonia between birth to 12 months of age by vitamin D status at birth. (b) Kaplan-Meier survival curve of the first episodes of pneumonia between 6 to 12 months of age by vitamin D status at six months of age.

contribute to the variation in findings [44–46]. Evaluation of the impact of low-dose daily or weekly vitamin D supplementation on severe pneumonia in children with profound deficiency has been recommended [41, 47].

Our study showed that a combination of more time spent outdoors with ambient UVB during pregnancy was associated with a reduced risk of pneumonia during infancy. Safe sun exposure practice with repeated low-dose summer sunlight has been associated with vitamin D sufficiency without causing significant DNA damage [48]. Accordingly, native populations in sun-abundant countries such as Indonesia where the majority of skin types range from medium to dark brown could benefit from sunlight but the optimum and safe duration and time of the day under the sun need further elucidation.

It is biologically plausible that vitamin D deficiency may be associated with susceptibility to respiratory infection. Laboratory studies demonstrate that vitamin D plays a direct role in innate immune responses regulating the expression of human antimicrobial peptides (cathelicidin and human beta-defensin 2) for pathogen elimination and reduction of viral replication while in the adaptive immune system, vitamin D suppresses lymphocytes T cells expression to further avoid immune pathology that can injure the lung tissues [25, 49]. However, to what extent the immunomodulatory roles of vitamin D influence the clinical manifestation and severity of respiratory diseases remain uncertain with ongoing controversy about the possible effect of vitamin D on ARI and the potential clinical and public health impact [50].

The strengths of our study include the use of active and passive ARI surveillance with a short assessment period and the use of the standardised WHO case definition for the diagnosis of pneumonia. This has been lacking in previous cohort studies examining an association between vitamin D and ARI. Despite careful, frequent, and thorough case-ascertainment, it is possible that we may have missed some pneumonia episodes and capture less severe cases. If missed episodes occurred it is likely that these would have been random, and independent of the infants' vitamin D status.

## Conclusion

ARI and pneumonia were common in this large cohort of Indonesian infants, among whom vitamin D deficiency was highly prevalent at birth. Vitamin D deficiency was associated with the incidence of ARI but not of pneumonia in the first year of life. Minimising vitamin D deficiency for infants in this setting has potential benefits to reduce ARI incidence and ARI-related healthcare utilisation, which needs to be determined in a therapeutic trial.

## Supporting information

**S1 File. Study questionnaire and case report files.**
(PDF)

## Acknowledgments

We thank all infants and parents who participated to this study, all study site staffs from Puskesmas, midwife private practice, health office (Dinas kesehatan kota Yogyakarta and Kulon Progo), and district hospitals in Kota Yogyakarta and Kulon Progo who helped with the study recruitment. Lastly, we especially thank all IPADS research assistants who assisted with data collection and study conduct.

## Author Contributions

**Conceptualization:** Vicka Oktaria, Margaret Danchin, Rina Triasih, Julie E. Bines, Anne-Louise Ponsonby, Stephen M. Graham.

**Data curation:** Vicka Oktaria.

**Formal analysis:** Vicka Oktaria, Michael W. Clarke.

**Funding acquisition:** Vicka Oktaria, Margaret Danchin, Yati Soenarto, Stephen M. Graham.

**Investigation:** Vicka Oktaria, Margaret Danchin, Rina Triasih, Yati Soenarto, Julie E. Bines, Stephen M. Graham.

**Methodology:** Vicka Oktaria, Margaret Danchin, Rina Triasih, Julie E. Bines, Anne-Louise Ponsonby, Stephen M. Graham.

**Resources:** Vicka Oktaria, Margaret Danchin, Yati Soenarto, Stephen M. Graham.

**Supervision:** Vicka Oktaria, Margaret Danchin, Rina Triasih, Yati Soenarto, Julie E. Bines, Anne-Louise Ponsonby, Stephen M. Graham.

**Validation:** Vicka Oktaria, Margaret Danchin, Michael W. Clarke, Stephen M. Graham.

**Writing – original draft:** Vicka Oktaria.

**Writing – review & editing:** Vicka Oktaria, Margaret Danchin, Rina Triasih, Yati Soenarto, Julie E. Bines, Anne-Louise Ponsonby, Michael W. Clarke, Stephen M. Graham.

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
