## [Decision Letter · Decision Letter 0]

30 Dec 2020

PONE-D-20-33592

The incidence of acute respiratory infection in Indonesian infants and association with vitamin D deficiency

PLOS ONE

Dear Dr. Oktaria,

Thank you for submitting your manuscript to PLOS ONE. After careful consideration, we feel that it has merit but does not fully meet PLOS ONE’s publication criteria as it currently stands. Therefore, we invite you to submit a revised version of the manuscript that addresses the points raised during the review process.

I must congratulate the authors for a well-performed study that enlighten the risk of vitamin D deficiency in childhood.

In addition to the issues raised by the reviewers, please provide a little more details about the performance of the vitamin D assay (CV, internal standard, QC data). I appreciate that they have use MS method rather than immunological methods, but is both D2 and D3 measured. Also, please consider the definition of deficiency 25/50/75 nmol/L, is that cast in bronze? Considering that you have measured with a relatively reliable method a quantitative test rather than categorical (deficiency or not) would have been preferable.

You have considered maternal sun exposition as D3 source, I assume that season is less likely to influence this as Indonesia is close to equator, but is social factors involved (clothing?). Also, please provide a note on maritime diet.

We look forward to receiving your revised manuscript.

Kind regards,

Pal Bela Szecsi, M.D. D.M.Sci.

Academic Editor

PLOS ONE

Journal Requirements:

2. Please provide additional details regarding participant consent. In the ethics statement in the Methods and online submission information, please ensure that you have specified whether consent was informed.

3. Please include additional information regarding the structured questionnaires and telephone interview guides used in the study and ensure that you have provided sufficient details that others could replicate the analyses.

For instance, if you developed the questionnaires/guides as part of this study and they are not under a copyright more restrictive than CC-BY, please include a copy, in both the original language and English, as Supporting Information.

4. Please state the date range of patient recruitment (month and year).

5. Please list the nine Primary Healthcare Centres and five private practice clinics where patients were recruited from.

Reviewers' comments:

Reviewer's Responses to Questions

**Comments to the Author**

1. Is the manuscript technically sound, and do the data support the conclusions?

Reviewer #1: Partly

Reviewer #2: Yes

2. Has the statistical analysis been performed appropriately and rigorously? 

Reviewer #1: Yes

Reviewer #2: Yes

3. Have the authors made all data underlying the findings in their manuscript fully available?

Reviewer #1: Yes

Reviewer #2: No

4. Is the manuscript presented in an intelligible fashion and written in standard English?

Reviewer #1: Yes

Reviewer #2: Yes

5. Review Comments to the Author

Reviewer #1: Oktaria et al. tried to investigate the vitamin-D levels of the ARI affected infants and the possible correlation of this vitamin with the incidence of ARI. Eventually they concluded that Vitamin D deficiency was associated with the incidence of ARI but not of pneumonia in the first year of life but adequate maternal sun exposure during pregnancy was associated with a trend to less frequent ARI and pneumonia in Indonesian infants.

My observations-

1. Here the authors mentioned that similar prospective cohort community-based cohort studies that measure actual pneumonia incidence are limited. However, if they can compare or discuss results from any other Indonesian or South Asian studies about ARI in infants especially with regards to immunization/breastfeeding practices/socioeconomic status, it would make the research paper even better and add more credibility to the results.

2. Despite the great efforts of the authors, the data obtained in this study lacks innovation, it is recommended that the paper be submitted to the local journals.

Reviewer #2: Uploaded as an attachment.

Please note data and scripts are not available to the reviewers.

##########################################################

##########################################################

6. PLOS authors have the option to publish the peer review history of their article (what does this mean?). If published, this will include your full peer review and any attached files.

Reviewer #1: **Yes: **Md. Rabiul Islam

Reviewer #2: **Yes: **Antonio J Berlanga-Taylor

---

## [Author Response · Author response to Decision Letter 0]

8 Feb 2021

The incidence of acute respiratory infection in Indonesian infants and association with vitamin D deficiency

Submission ID: PONE-D-20-33592

Response to reviewers: 

Editor

I must congratulate the authors for a well-performed study that enlighten the risk of vitamin D deficiency in childhood. In addition to the issues raised by the reviewers, please provide a little more detail about the performance of the vitamin D assay (CV, internal standard, QC data). I appreciate that they have use MS method rather than immunological methods but is both D2 and D3 measured. Also, please consider the definition of deficiency 25/50/75 nmol/L, is that cast in bronze? Considering that you have measured with a relatively reliable method a quantitative test rather than categorical (deficiency or not) would have been preferable.

Author response:

Many thanks for your positive comment. We have added some information on the Coefficient of Variability (CV), internal standard and Quality control data as stated in the paper – Methods, Page 9 -10, line 197 - 209. The assay gives excellent separation of epi-25(OH)D3, and 25(OH)D2 from 25(OH)D3, has excellent precision with an intra-assay CV of 0.5 % at 74 nmol/L for 25(OH)D3. 

In our recently published paper, we categorised vitamin D deficiency using different cut-offs [Oktaria V, et al. (2020) The prevalence and determinants of vitamin D deficiency in Indonesian infants at birth and six months of age. PLoS ONE 15(10): e0239603.] We have run analysis in the current manuscript using different cut-offs of <25 and <75 nmol/L but this did not substantially change the results in comparison with a cut-off of <50 nmol/L. We therefore use < 50 nmol/L for the definition of deficiency as it is the most commonly reported in measuring the association between respiratory infections and vitamin D deficiency. 

Change in the manuscript: Page 9 -10, line 197 - 209

“Sera from cord blood and venous blood were assayed at Metabolomics Australia, Centre for Microscopy, Characterisation, and Analysis, University of Western Australia. The total serum 25-hydroxyvitamin D concentrations were measured using liquid chromatography-tandem mass spectrometry, the current gold standard for vitamin D measurement, which has shown excellent agreement with Center for Disease Control and Prevention’s 25-hydroxyvitamin D inaugural Vitamin D standardisation program (r2 =0.99) (31). The analytical sensitivity (or limit of detection) and the functional sensitivity (or limit of quantitation) of the assay for 25-hydroxyvitamin D3 is 0.5 nM and 2 nmol/L respectively. 

Commercial quality controls are assayed at the beginning and end of each batch, with intra-assay Coefficient of variability’s (CV’s) for all three metabolites <5%, with inter-assay CV’s being <10%. Internal standard peak areas are monitored for each batch and are <10% within each run. We defined vitamin D deficiency as a serum 25-hydroxyvitamin D3 level of <50 nmol/L.”

You have considered maternal sun exposition as D3 source, I assume that season is less likely to influence this as Indonesia is close to equator, but is social factors involved (clothing?). Also, please provide a note on maritime diet.

Author response:

As you suggest, Indonesia has limited seasonal fluctuation of sun exposure. During our study period 2015 - 2017, the median of sun exposure was 266 (IQR 242 – 290) mW/m2/day. We have included discussion about maternal clothing and infant diet in a separate publication where we reported that “only 5% of our study infants had egg yolk in the first six months, mainly those who were living in the rural area, and even less had had fish or red meats.”. [Oktaria V, et al. (2020) The prevalence and determinants of vitamin D deficiency in Indonesian infants at birth and six months of age. PLoS ONE 15(10): e0239603.] We collected information on maritime diet (seafood intake) in infants but unfortunately not in mothers. 

We also reported in the publication that time spent outdoor during pregnancy was associated with cord blood vitamin D concentration. Clothing (represented by skin sun exposure score) was associated with vitamin D concentration in infant but not in mother. However, other study from Indonesia reported a significant association between vitamin D status and maternal clothing [Judistiani RTD, et al. Optimizing ultra- violet B radiation exposure to prevent vitamin D deficiency among pregnant women in the tropical zone: report from cohort study on vitamin D status and its impact during pregnancy in Indonesia. BMC Pregnancy Childbirth. 2019; 19(1):209. https://doi.org/10.1186/s12884-019-2306-7 PMID: 31226954]. 

Change made in the manuscript: None 

Additional requirement from the journals

Author response:

We have followed the recommended guideline.

2. Please provide additional details regarding participant consent. In the ethics statement in the Methods and online submission information, please ensure that you have specified whether consent was informed.

Author response:

We have now amended the sentences in the Methods, Page 7-8, Line 156 – 159

Change made in the manuscript:

“Informed consent was obtained prior to delivery and included the intention to actively follow infants for 12 months, and to collect cord blood at delivery and a venous blood sample at six months of age. Written informed consent was reaffirmed after delivery of a live birth.” 

3. Please include additional information regarding the structured questionnaires and telephone interview guides used in the study and ensure that you have provided sufficient details that others could replicate the analyses.

For instance, if you developed the questionnaires/guides as part of this study and they are not under a copyright more restrictive than CC-BY, please include a copy, in both the original language and English, as Supporting Information.

Author response

Yes, we have included the questionnaire as supporting information (S1 Files)

4. Please state the date range of patient recruitment (month and year).

Author response: 

Dates now added in Abstract Page 3, Line 44-45 and Result section Page 11, Line 239 -240

Change made in the manuscript:

Between 23 December 2015 and 31 December 2017, we recruited 422 participants with 95% (400/422) of infants retained for the full twelve months of follow-up. 

5. Please list the nine Primary Healthcare Centres and five private practice clinics where patients were recruited from.

Author response :

We have now listed the clinics in Page 7, line 143 – 146

Change made in the manuscript:

A prospective birth cohort study was conducted in nine Primary Healthcare Centres (Jetis, Tegalrejo, Gedongtengen, Gondokusuman 2, Mantrijeron, Gondomanan, Wates, Sentolo, Pengasih 1) and five private practice clinics (Sri Suharti, Suwarti, Sulalita, Sri Suyantiningsih, and Sri Esthini) located in two districts in Yogyakarta province, Indonesia. 

Reviewer 1

1. Here the authors mentioned that similar prospective cohort community-based cohort studies that measure actual pneumonia incidence are limited. However, if they can compare or discuss results from any other Indonesian or South Asian studies about ARI in infants especially with regards to immunization/breastfeeding practices/socioeconomic status, it would make the research paper even better and add more credibility to the results.

Author response:

Thank you for highlighting this point. Prospective studies of ARI incidence in infants in Indonesia or the region are limited. The additional challenge for comparison is that other studies utilize variable definitions of ARI, URTI and LRTI as well as methods of ARI case ascertainment. Standardization of ARI criteria are important because application of different definitions to the same cohort could substantially change the reported burden. 

Most “current” data are from epidemiological modelling. For example, Rudan I et al. conducted periodical epidemiological modeling for pneumonia burden in 2004, 2008 and 2013 and showed little variation in the included studies from one period to the other. A recognised limitation for epidemiological analysis was the “slow growth” of prospective studies on pneumonia incidence. This meant that the statistical predictive model was partly influenced by studies that were published prior to 2000, and yet the incidence may have been overestimated because the epidemiology is likely changing. 

We have now cited important publications from the region that link known risk factors for ARI such as exclusive breastfeeding, poverty and ARI in Introduction section, Page 6, Line 118 - 121 

Change made in the manuscript:

ARI risk factors in Indonesia are similar to the global population. These include poverty, indoor pollution, suboptimum exclusive breastfeeding and poor nutritional status.[5, 14] Nutritional deficiency, including both macro and micronutrient deficiencies, is common in Indonesian children [15, 16].

2. Despite the great efforts of the authors, the data obtained in this study lacks innovation, it is recommended that the paper be submitted to the local journals.

Author response:

Thank you very much for the suggestions. We believe that our study results could provide some valuable information for other countries that shared similar geographic, cultural and religious backgrounds with Indonesia. The wide range of PLOS ONE readers would enhance the dissemination of our study results. 

Change made in the manuscript: None 

Reviewer 2

Abstract 

1. Lines 62-63: Conclusion: Minimising vitamin D deficiency at birth such as by supplementation or safe sun exposure has the potential to reduce ARI incidence in infants in this setting.” >Supplementation of mothers during pregnancy? 

Author response:

We have amended the sentences

Change made in the manuscript: Page 4, line 62 – 64

ARI, pneumonia, and vitamin D deficiency at birth were common in Indonesian infants. Minimising vitamin D deficiency at birth such as by supplementation of mothers or safe sun exposure during pregnancy has the potential to reduce ARI incidence in infants in this setting. 

Introduction

2. The introduction is informative, presents some background knowledge on the subject and literature. However, I would suggest providing some more specific information on the Indonesian population e.g.:

a. What is the prevalence of ARI in Indonesia, how many infants diagnosed each year, mortality rate, etc.

Author response:

We have added such information - Page 5, Line 96 - 100

Change made in the manuscript:

In the same year, the total number of estimated pneumonia episodes in Indonesian children aged younger than five years old were 3,196,000 episodes (2,447,000 to 3,666,000 episodes), with an incidence rate of ~ 300 cases per 1000 population.(1, 3) The total number of estimated deaths reported in that year was 15,250 deaths (9,900 to 20,124 deaths) (2, 5-7). 

b. What are some known risk factors of ARIs and how does this relate to the Indonesian population?

Author response:

We have added more information - Page 6, Line 118 - 121 

Change made in the manuscript:

ARI risk factors in Indonesia are similar to the global population. These include poverty, indoor pollution, suboptimum exclusive breastfeeding and poor nutritional status.[5, 14] Nutritional deficiency, including both macro and micronutrient deficiencies, is common in Indonesian children [15, 16].

c. Are ARIs the leading cause of death in Indonesian children? What about different age groups?

Author response:

We have added further information - Page 5, Line 96 - 100

Change made in the manuscript:

In the same year, the total number of estimated pneumonia episodes in Indonesian children aged younger than five years old were 3,196,000 episodes (2,447,000 to 3,666,000 episodes), with an incidence rate of ~ 300 cases per 1000 population.(1, 3) The total number of estimated deaths reported in that year was 15,250 deaths (9,900 to 20,124 deaths) (2, 5-7). 

d. Also, some additional background information is required regarding pneumonia in children.

Author response:

We have added some information as described above in the response to query 2 a-c

e. What about prevalence of vitamin D deficiency in Indonesian children?

Author response:

We have added information: Page 7, line 134 – 135

Change made in the manuscript:

The prevalence of vitamin D deficiency in Indonesia children aged 2-5 years old was 44% and newborns have been reported at high risk of vitamin D deficiency (28, 29). 

3. Line 109-111: discussion of limited evidence currently reads a bit confusing, would recommend some re-wording.

Author response:

We have amended the sentences - Page 6, line 127 – 130.

Change made in the manuscript:

There is biological plausibility for this association given the immunomodulatory effects of vitamin D deficiency (24). However, a limited number of studies on vitamin D supplementation in children with ARI have been published to date and evidence of a convincing association with prevention or treatment of outcomes is still lacking. (16, 25). 

4. Lines 85-91 >After line 86 I would suggest having a sentence to introduce the reader that ARI can affect either upper respiratory system or lower respiratory system and then go into details of each URTI, LRTI, and pneumonia more specifically. 

Author response:

We have amended the sentences - Page 5, line 87 – 91

Change made in the manuscript:

Acute respiratory infection (ARI) is the leading cause of disease and death in young children (<5 years) globally (1, 2). ARI incorporates a wide range of respiratory illnesses, from mild to life-threatening, that are classified based on location of infection in relation to the vocal cords into: upper respiratory tract infection (URTI) such as rhinitis, nasopharyngitis, tonsillitis, or epiglottitis; and lower respiratory tract infection (LRTI) such as pneumonia, bronchiolitis or croup (3). Upper respiratory tract infection (URTI) is a very common outpatient presentation associated with low mortality but significant health systems costs (1, 4) 

5. Line 95 >Please provide more details regarding the Haemophilus influenzae type b vaccine in the introduction e.g. When was it implemented? Does it prevent pneumonia?

Author response:

We have amended the sentences - Page 5-6, Line 107 - 114

Change made in the manuscript:

The major recognised bacterial pathogens causing pneumonia, influencing global efforts to prevent pneumonia-related deaths through case management and vaccines have been Streptococcus pneumoniae and Haemophilus influenzae type b (Hib) (2). Since the Hib vaccine was included in the routine immunisation schedule in 2013, the prevalence of Hib carriage has been shown to decline to 0% compared to 5% as reported in 1998 (10, 11). With socioeconomic development and the introduction of bacterial conjugate vaccines, the epidemiology is changing with a relatively larger proportion of ARI due to viruses and a lower mortality attributed to ARI (12). 

6. Page 5 lines 106-108 “Subclinical vitamin D deficiency (serum vitamin D <50 nmol/L) is far more common and has…” And Line 180: We defined vitamin D deficiency as a serum 25-hydroxyvitamin D level of <50nmol/L.

a. It would be good to provide a reference for these statements. 

b. Add “far more common in infants” if this is this what the sentence implies. 

Author response:

We have amended the sentences and added reference for the statement - Page 6, line 124 - 127

Change made in the manuscript:

Subclinical vitamin D deficiency (serum vitamin D <50 nmol/L) is far more common than clinical rickets in young children (20-22) and has also been associated with ARI, including increased risk of URTI, LRTI, and LRTI hospitalisation (17, 18, 23). 

7. Page 6 lines 108-111 “However, evidence of a causal relationship is inconclusive and limited studies of vitamin D supplementation have not yet demonstrated benefit for prevention or treatment outcomes [9, 15]”

a. I assume the abovementioned limited studies are in children, or do they include adults as well? 

b. Could rephrase to: A limited number of studies on vitamin D supplementation in children with ARI have been published to date and evidence of a convincing association with prevention or treatment of outcomes is still lacking. 

Author response:

Yes, it is correct that we highlighted the limited evidence in children. We have rephrased as suggested. Page 6, Line 127 - 130

Change made in the manuscript:

There is biological plausibility for this association given the immunomodulatory effects of vitamin D deficiency (24). However, a limited number of studies on vitamin D supplementation in children with ARI have been published to date and evidence of a convincing association with prevention or treatment of outcomes is still lacking. (16, 25). 

8. Line 129 Study procedures: Any exclusion criteria that should be mentioned? Both parents were interviewed or only mother?

Author response:

We primarily interviewed mother

We have amended the sentences and added the exclusion criteria - Page 7, Line 154 - 156

Change made in the manuscript:

Inclusion criteria included expected delivery in one of the study sites without intention to leave the area within 12 months. Parents who did not provide written informed consent for their child’s participation or who did not have phone access were not included. 

9. Lines 146-147 “ if the participants reported cough for more than three days to obtain the respiratory rate” >Do the participants imply the infants? And whenever the word participants is used I would suggest to use infants for consistency. 

Author response:

Yes, participant imply infants. We have amended as suggested

Change made in the manuscript:

Replacement of participants into infants throughout the draft have been done as appropriate

10. Line 150 according to WHO guidelines > Reference?

Author response:

We have added a reference to the related sentence. 

Methods

11. Line 144: It would be good to have a description of what constituted “respiratory symptoms”. It is a bit unclear if a classification of “ARI non-pneumonia episode” required a home visit to be considered for the study (e.g. cough over 3 days?) and how these episodes were categorized.

Author response:

We have clarified the definition for ARI non pneumonia. Page 9. Line 185 - 187

The initial symptom of ARI pneumonia and non-pneumonia could be similar. We performed home visit and maintained regular contacts during symptomatic period to monitor if the symptoms worsening. Categorisation by WHO criteria for pneumonia was used and decision for the final diagnosis were discussed by the study doctor team led by paediatric respiratory experts. 

Change made in the manuscript:

ARI non-pneumonia episode was defined for respiratory symptoms (including runny nose, fever, sore throat and/or cough) without fast breathing or chest indrawing. 

Statistical analysis 

12. Which confounders were considered for the Cox Regression, please provide more details.

Author response:

We have listed the adjusted confounders for each evaluated association below Table 5. Confounding was determined: if the difference between the Odd Ratio (OR) or Hazard Ratio (HR) in the univariate and multivariate models changed by more than 10% or based on a priori knowledge.

13. I can see that certain sensitivity analyses were conducted e.g. using a lower cut-off for vitamin D (level of <25) but these are not measured in methods. It would be good to mention these in the statistical analysis.

Author response:

We have added sensitivity analysis in the method section - Page 10, Line 228 - 230

Change made in the manuscript:

A sensitivity analysis using a lower cut off vitamin D level of <25 nmol/L (severe vitamin D deficiency) and a higher cut off vitamin D level of < 75 nmol/L was performed. 

14. Was the proportional hazards assumption for Cox proportional hazards models tested?

Author response:

We have now added the Kaplan-Meier plots in the draft (Fig 2) and the graphs show parallel curves that crossing at some points. The hazard rates were not different between those with and without vitamin D deficiency at birth and at six months of age.

15. I don’t see any apparent risk of immortal person time according to classification of exposures and outcomes.

Author response:

We do not quite understand this query. Would the reviewer please kindly explain in more detail?

16. Line 171-172: It is possible that seasonality impacts maternal sun exposure, unclear if this is accounted for in the calculation of cumulative composite estimate UVR (Wierzejska, R., Jarosz, M., Sawicki, W., Bachanek, M. and Siuba-Strzelińska, M., 2017. Vitamin D concentration in maternal and umbilical cord blood by season. International journal of environmental research and public health, 14(10), p.1121.)

Author response:

Indonesia has year round sun exposure with limited seasonal fluctuation. During our study period 2015 - 2017, the median of sun exposure was 266 (IQR 242 – 290) mW/m2/day. 

Maternal sun exposures are greatly influenced by clothing preference that was strongly affected by religious beliefs (Muslim population). Muslim female wear covering clothes (hijab) regardless the season/whether condition. Our recent published work reported that majority (90%) of our participating infants were vitamin D deficiency at birth with longer time outdoors during pregnancy and maternal multivitamin and maternal multivitamin intake containing vitamin D during pregnancy were protective against vitamin D deficiency at birth [Oktaria V, et al. (2020) The prevalence and determinants of vitamin D deficiency in Indonesian infants at birth and six months of age. PLoS ONE 15(10): e0239603.]

17. Data are not available to reviewers. Please upload these and deposit somewhere accessible.

Author response:

We have now deposited our data in Figshare, DOI: 10.6084/m9.figshare.13726525

18. Analyses scripts are not available either. Please upload these as well and deposit somewhere accessible.

Author response:

We have now deposited our do file analysis in Figshare, DOI: 10.6084/m9.figshare.13726525

Results

19. “Between December 2015 and December 2017, we recruited 422 participants with 95% 210 (400/422) of infants retained for the full twelve months of follow-up.”

a. Participant is mothers or infants?

b. Any important information for those lost to follow up?

Author response:

 Participants are infants and we have amended the wording throughout the draft

We have listed all information in Figure 1. The time period of observation for each infant was determined as the period in days from the date of birth until the date of the last contact made (final follow-up, prior to lost to follow-up or date of death). All ARI outcomes reported prior to loss of contact were analyzed. 

20. Please show plots as well as tables.

Author response:

We have reported plots on the prevalence of infant vitamin D deficiency and correlation between cord blood and six-month blood vitamin D level in a separate publication [Oktaria V, et al. (2020) The prevalence and determinants of vitamin D deficiency in Indonesian infants at birth and six months of age. PLoS ONE 15(10): e0239603.]. 

We have now added the Kaplan Meier plots for association between vitamin D status and pneumonia in this draft (Fig 2).

Change made in the manuscript:

Figure 2 Time to develop first episode of pneumonia by vitamin D status at birth and 6 months of age

21. Lines 224-225 The demographics characteristics of those with vitamin D samples at birth and six months of 225 age were representative of the entire cohort (Table 1).

a. Specify “those”, infants?

b. Change participants to infants in Table 1

Author response:

Participants are infants and we have amended the wording throughout the draft. We have edited the statement on Page 11, Line 254 - 255

Change made in the manuscript:

The demographics characteristics of infants with vitamin D samples at birth and six months of age were representative of the entire cohort (Table 1). 

Discussion

22. Line 349-350: A major potential confounder exists in the assessment of maternal sun exposure – since this is likely to be higher in more rural areas and infections may be more prevalent in urban ones. Was there any measure available for urbanicity? Household crowding was considered but it is not reflective of wider environment.

Author response:

The inclusion of urbanicity did not materially change the association between maternal sun exposure and pneumonia and did not change the strength of the association in the multivariate models. 

Change made in the manuscript: None

23. Line 361: Vitamin D supplementation is discussed, was there any data collection on maternal use of supplements?

Author response:

Maternal vitamin D supplementation in pregnant women was not standard recommendation in our setting. A minority (7%) of mothers reported taking a multivitamin supplement that contained vitamin D during pregnancy with 50% reporting a daily vitamin D intake below 400 IU, the minimum recommended daily dose in pregnancy. Due to very low proportion of infants whose mothers had taken multivitamin supplement that contained vitamin D, we were unable to do further analysis. 

24. It seems a bit odd to me that deficiency according to cord blood would be so high, is this normal? I couldn’t find similar figures through a quick literature search.

Author response:

A similar study from Thailand demonstrated similar high prevalence of vitamin D deficiency in newborn (89.3%) [Ariyawatkul, K. and P. Lersbuasin (2018). "Prevalence of vitamin D deficiency in cord blood of newborns and the association with maternal vitamin D status." Eur J Pediatr 177(10): 1541-1545]. 

It is perhaps not surprising as a high prevalence of vitamin D deficiency was also reported in Indonesian mothers in pregnancy which half of them wore veils and covering clothes (80.4%). [Judistiani RTD, et al. Optimizing ultraviolet B radiation exposure to prevent vitamin D deficiency among pregnant women in the tropical zone: report from cohort study on vitamin D status and its impact during pregnancy in Indonesia. BMC Pregnancy Childbirth. 2019; 19(1):209. https://doi.org/10.1186/s12884-019-2306-7 PMID: 31226954].

---

## [Decision Letter · Decision Letter 1]

4 Mar 2021

The incidence of acute respiratory infection in Indonesian infants and association with vitamin D deficiency

PONE-D-20-33592R1

Dear Dr. Oktaria,

We’re pleased to inform you that your manuscript has been judged scientifically suitable for publication and will be formally accepted for publication once it meets all outstanding technical requirements.

Kind regards,

Pal Bela Szecsi, M.D. D.M.Sci.

Academic Editor

PLOS ONE

Additional Editor Comments (optional):

Reviewers' comments:

Reviewer's Responses to Questions

**Comments to the Author**

1. If the authors have adequately addressed your comments raised in a previous round of review and you feel that this manuscript is now acceptable for publication, you may indicate that here to bypass the “Comments to the Author” section, enter your conflict of interest statement in the “Confidential to Editor” section, and submit your "Accept" recommendation.

Reviewer #1: All comments have been addressed

Reviewer #2: All comments have been addressed

2. Is the manuscript technically sound, and do the data support the conclusions?

Reviewer #1: Partly

Reviewer #2: Yes

3. Has the statistical analysis been performed appropriately and rigorously? 

Reviewer #1: Yes

Reviewer #2: Yes

4. Have the authors made all data underlying the findings in their manuscript fully available?

Reviewer #1: Yes

Reviewer #2: Yes

5. Is the manuscript presented in an intelligible fashion and written in standard English?

Reviewer #1: Yes

Reviewer #2: Yes

6. Review Comments to the Author

Reviewer #1: Thanks to the authors for addressing my observations. The other reviewers noted several issues along with editorial comments and many changes have been made. Indeed, the paper is improved and I recommend publication.

Reviewer #2: No further comments from our side. If external replication confirms results it would certainly have important public health implications with potential for changing national policy. I hope authors can follow-up in a separate study and/or collaborate with a group in a different city/province. Many thanks for the detailed response and congratulations on an interesting paper.

7. PLOS authors have the option to publish the peer review history of their article (what does this mean?). If published, this will include your full peer review and any attached files.

Reviewer #1: **Yes: **Md. Rabiul Islam, Assistant Professor, Department of Pharmacy, University of Asia Pacific, Dhaka, Bangladesh

Reviewer #2: **Yes: **Antonio Berlanga

---

## [Editor Report · Acceptance letter]

15 Mar 2021

PONE-D-20-33592R1 

The incidence of acute respiratory infection in Indonesian infants and association with vitamin D deficiency 

Dear Dr. Oktaria:

I'm pleased to inform you that your manuscript has been deemed suitable for publication in PLOS ONE. Congratulations! Your manuscript is now with our production department. 

Kind regards, 

on behalf of

Dr. Pal Bela Szecsi 

Academic Editor

PLOS ONE